# Connexin 43 Controls the Astrocyte Immunoregulatory Phenotype

**DOI:** 10.3390/brainsci8040050

**Published:** 2018-03-22

**Authors:** Anne-Cécile Boulay, Alice Gilbert, Vanessa Oliveira Moreira, Corinne Blugeon, Sandrine Perrin, Juliette Pouch, Stéphane Le Crom, Bertrand Ducos, Martine Cohen-Salmon

**Affiliations:** 1Collège de France, Center for Interdisciplinary Research in Biology (CIRB)/Centre National de la Recherche Scientifique CNRS, Unité Mixte de Recherche 7241/Institut National de la Santé et de la Recherche Médicale INSERM, U1050/75231 Paris CEDEX 05, France; anne-cecile.boulay@college-de-france.fr (A.-C.B.); alice.gilbert@college-de-france.fr (A.G.); vanessa.oliveira-moreira@college-de-france.fr (V.O.M.); 2Paris Science Lettre Research University, 75005 Paris, France; 3Ecole Normale Supérieure, Institut de Biologie de l’ENS, IBENS, Genomic Facility, Institut de Biologie de l’Ecole Normale Supérieure (IBENS), Ecole Normale Supérieure, CNRS, INSERM, PSL Université Paris, 75005 Paris, France; blugeon@biologie.ens.fr (C.B.); sandrine.perrin7@free.fr (S.P.); juliette.hamroune@inserm.fr (J.P.); lecrom@biologie.ens.fr (S.L.C.); bertrand.ducos@lps.ens.fr (B.D.); 4Laboratoire de Physique Statistique-ENS CNRS UMR 8550, 75005 Paris, France; 5PSL Research University, Université Paris Diderot Sorbonne Paris-Cité, Sorbonne Universités UPMC Univ Paris 06, CNRS, 75005 Paris, France; 6High Throughput qPCR Core Facility, IBENS, 46 rue d’Ulm, 75005 Paris, France

**Keywords:** astrocyte, connexin 43, immunity, inflammation, ribosomal-bound transcriptome

## Abstract

Astrocytes are the most abundant glial cells of the central nervous system and have recently been recognized as crucial in the regulation of brain immunity. In most neuropathological conditions, astrocytes are prone to a radical phenotypical change called reactivity, which plays a key role in astrocyte contribution to neuroinflammation. However, how astrocytes regulate brain immunity in healthy conditions is an understudied question. One of the astroglial molecule involved in these regulations might be Connexin 43 (Cx43), a gap junction protein highly enriched in astrocyte perivascular endfeet-terminated processes forming the glia limitans. Indeed, Cx43 deletion in astrocytes (Cx43KO) promotes a continuous immune recruitment and an autoimmune response against an astrocyte protein, without inducing any brain lesion. To investigate the molecular basis of this unique immune response, we characterized the polysomal transcriptome of hippocampal astrocytes deleted for Cx43. Our results demonstrate that, in the absence of Cx43, astrocytes adopt an atypical reactive status with no change in most canonical astrogliosis markers, but with an upregulation of molecules promoting immune recruitment, complement activation as well as anti-inflammatory processes. Intriguingly, while several of these upregulated transcriptional events suggested an activation of the γ-interferon pathway, no increase in this cytokine or activation of related signaling pathways were found in Cx43KO. Finally, deletion of astroglial Cx43 was associated with the upregulation of several angiogenic factors, consistent with an increase in microvascular density in Cx43KO brains. Collectively, these results strongly suggest that Cx43 controls immunoregulatory and angiogenic properties of astrocytes.

## 1. Introduction

The brain is subjected to a constant immune surveillance, which allows for a rapid and efficient reaction against insults and infections. Nevertheless, in healthy conditions and in contrast to most peripheral organs, entry and survival in the brain parenchyma of immune cells circulating in the blood or present in the cerebrospinal fluid (CSF) filling perivascular spaces are very restricted [1]. In particular, T cells rarely penetrate the brain in absence of antigenic triggering. Such unique and quiescent environment relies on anatomical barriers: the epithelial cells of the choroid plexus forming the blood-CSF barrier and the endothelial cells of parenchymal blood vessels, the blood-brain barrier (BBB). These cells are indeed sealed by tight junctions (TJs), which impede the passage of CSF and blood components. In addition, under physiological conditions, brain endothelial cells are unable to interact with circulating leukocytes as they lack specific surface receptors and are equipped with efflux transporters and enzymes able to restrain the passage of blood macromolecules. This endothelial barrier greatly relies on perivascular neural cells, the pericytes and the astrocytes, which induce endothelial barrier properties during the development and maintain them throughout life [2,3]. Astrocytes, the most abundant glial cells in the brain, are ramified cells with processes terminated by a perivascular compartment called endfeet covered with a basal lamina. Astrocyte endfeet entirely sheath the brain vessels, either in direct contact, at the microvascular level, or separated from the vessel walls by perivascular spaces filled with CSF, at the level of large penetrating vessels. Such continuous astrocyte perivascular coverage constitutes the *glia limitans* (GL). Although devoid of TJ, the GL isolates the vascular compartment from the parenchyma and is thus considered as a second barrier to immune cell penetration [4]. This role is particularly evident in physiopathological conditions such as inflammation, upon which astrocytes become reactive, changing their morphology and molecular profile [5,6,7], in a way that strongly modifies immune quiescence and BBB integrity [2,8]. Although deep efforts have been put to understand how reactive astrocytes behave in physiopathological conditions, the way they contribute to the physiological homeostatic immune equilibrium in the brain has been poorly explored.

The GL is equipped with a specific molecular repertoire including proteins such as the water channel Aquaporin 4 (Aqp4) and the potassium channel Kir4.1, which regulate the perivascular homeostasis, or the Dystroglycan complex linking basal lamina to the intracellular cytoskeleton [2,9]. Astrocyte endfeet forming the GL are also biochemically coupled by gap junctions, mainly composed of Connexin (Cx) 30 and 43 [10], as well as Cx26 in astrocytes facing the meninges [11]. At the plasma membrane, Cxs are grouped by six to form hemichannels permeable to ions and molecules of low molecular weight (1–1.2 kDa). Their assembly between two adjacent cells forms gap-type intercellular junctions, allowing cell-to-cell communication by diffusion. Besides their channel properties, Cxs display specific functions mainly related to their interaction with other intracellular proteins, such as signal transduction [12].

In a previous study, we showed that Cx43 might be one of the astroglial factor regulating the normal homeostasis between the brain and the immune system [13]. In absence of any pathogenic trigger, immune cells were recruited in the brain parenchyma of mice deleted for astroglial Cx43. They comprised B-lymphocytes and plasma cells, T-lymphocytes (CD4+ and CD8+), macrophages and neutrophils (Gr1+). This immune recruitment was associated with the development of an autoimmune humoral response against Vwa5a, an extracellular matrix protein expressed by astrocytes. However, no complement activation and no associated tissue lesion was observed in astroglial Cx43 deleted brain, but only a progressive weakening of the BBB [13]. How deletion of astroglial Cx43 lead to this atypical immune phenotype in the brain was not known. To answer this question, we analyzed the polysomal transcriptome of astrocyte deleted for Cx43.

## 2. Materials and Methods

### 2.1. Mice

Animals used in this study were kept in pathogen free conditions. The Cre-recombinase activity in the brain of Cx43^fl/fl^/hGFAP-Cre (Cx43KO) and Cx43^fl/fl^ (Cx43FL) mice was systematically tested before performing further experiments by revealing the β-galactosidase activity (Roche) [14]. In addition, since germ-line recombination were observed hGfap-Cre transgenic females offspring [15], only hGfap-Cre males were used for breeding. To obtain the astrocyte ribosome-bound transcriptome, Cx43KO and Cx43FL mice were bred with Tg (Aldh1l1-eGFP/Rpl10a) JD130Htz (MGI: 5496674) (Aldh1l1:L10a-eGFP) mice obtained from the laboratory of Dr. Nathaniel Heintz (Rockefeller University, New York, NY, USA). This mouse strain has been generated by BAC (bacterial artificial chromosome) transgenesis [16,17]. The Aldh1l1:L10a-eGFP allele was kept in a heterozygote state (see www.bactrap.org
*for the genotyping protocol*).

### 2.2. Study Approval

Experiments and techniques reported here complied with the ethical rules of the French agency for animal experimentation.

### 2.3. Antibodies

Primary antibodies: Rat monoclonal anti-Pecam-1/CD31 (Clone MEC 13.3, BD Pharmingen, San Jose, CA, USA); rabbit polyclonal anti-STAT1 (ab2415, Abcam, Cambridge, UK); mouse monoclonal anti-STAT1 pY701 (ab29045, Abcam, Cambridge, UK); rabbit polyclonal anti-NFκB/p65 (PA1–186, Thermo Fisher, Waltham, MA, USA); rabbit polyclonal anti-NFκB/p65 pThr435 (PA5–37724, ThermoFisher, Waltham, MA, USA); mouse monoclonal Anti-γ Interferon (ab133566, Abcam, Cambridge, UK).

Secondary antibodies: Alexa594-conjugated goat anti-rat antibodies (Pecam-1 immunolabeling on brain slices), Horseradish peroxidase (HRP)-conjugated goat anti-mouse and anti-rabbit (Western-blot).

### 2.4. Western-Blot 

Dissected hippocampi from 6 weeks-old animals were homogenized in Phosphate buffered saline (PBS) containing 2% SDS and EDTA-free Complete Protease Inhibitor (Roche, Basel, Switzerland), sonicated three times at 20 Hz (Vibra cell VCX130) and centrifuged 20 min at 10,000× *g* at 4 °C. Supernatants were boiled in Laemmli loading buffer. Protein content was measured using the Pierce 660 nm protein assay reagent (Thermo Fisher, Waltham, MA, USA). Equal amounts of proteins were separated by denaturing electrophoresis in 4–12% NuPAGE gradient gels (Thermo Fisher) and electrotransfered to nitrocellulose membranes. Membranes were analyzed as previously described [10]. HRP activity was visualized by ECL using Western Lightning plus enhanced chemoluminescence system (Perkin Elmer, Waltham, MA, USA). Chemoluminescence imaging was performed on a LAS4000 (Fujifilm, Minato-ku, Tokyo, Japan). GAPDH expression was used as a loading reference for γIFN expression. Total amount of STAT-1 and NFκB were used as a reference to determine proportion of phosphorylated proteins, pSTAT-1 and pNFκB respectively. All experiments were done in triplicates (*n* = 3). Cx43FL and Cx43KO expression levels were compared using the Mann-Whitney two-tailed test; ns stands for *p* > 0.05. Mean values are indicated ± the Standard Deviation (SD).

### 2.5. Translating Ribosome Affinity Purification (TRAP) in Astrocytes, RNA Sequencing and Analysis

Whole hippocampi of 6 weeks-old *Aldh1l1:L10a-eGFP* Cx43KO and *Aldhl1:L10a-eGFP Cx43FL mice* were used for each astrocyte polysome extraction (n = 3 libraries per genotype). Polysome extraction was performed following the protocol described in the bacTRAP project web site (www.bactrap.org) and by the authors of [18].

*Library preparation and Illumina sequencing*: 2 ng of total polysomal RNAs were amplified and converted to cDNA using NuGEN’s Ovation RNA-Seq kit. Following amplification, 1 µg of cDNA was fragmented to ~300 bps using Covaris S200. The remainder of the library preparation was done using 200 ng of cDNA following TruSeq RNA Sample Prep v2 kit from the End Repair step. Libraries were multiplexed by 3 on 2 flow cell lanes. A 50 bp read sequencing was performed on a HiSeq 1500 device (Illumina, San Diego, CA, USA). A mean of 50 ± 18 million passing Illumina quality filter reads was obtained for each sample.

*RNASeq bioinformatics analysis*: Analyses were performed using the Eoulsan pipeline version 2.0-alpha5 [19], including read filtering, mapping, alignment filtering, read quantification. Before mapping, reads ≤40 bases were removed, and reads with quality mean ≤30 were discarded. Reads were then aligned against the *Mus musculus* genome (mm10 version Ensembl 75) using STAR (version 2.4.0j, with default parameters) [20]. Alignments from reads matching more than once on the reference genome were removed. To compute gene expression, *Mus musculus* GFF3 genome annotation version mm10 version 75 from Ensembl database was used. All overlapping regions between alignments and referenced genes were counted using HTSeq-count 0.5.3 [21]. The RNASeq gene expression data and raw FASTQ files are available on the Gene Expression Omnibus (*GEO*) repository (www.ncbi.nlm.nih.gov/geo/) under accession number GSE110721.

*Differential gene expression analysis:* For each sample group (Cx43FL vs. Cx43KO), we assessed which genes displayed statistically significant differential expression using the Bioconductor package for differential analysis of count data (DESeq2 version 1.4.5) written for the R statistical programming environment. DESeq2 assumes the RNA-seq counts are distributed according to negative binomial distributions. It uses generalized linear modeling to test individual null hypotheses of a log_2_ fold changes of zero between conditions for each gene [19]. Lists of differentially expressed genes were further refined as follows: only events significantly different between genotypes (with a *p*-value < 0.05) were further considered; for genes upregulated in Cx43KO, only mRNAs with more than 50 reads in Cx43KO were selected; for genes downregulated in Cx43KO, only mRNAs with more than 50 reads in Cx43FL were selected;

*Gene Ontology (GO) analyses:* We used the text-mining Pathway Studio ResNet database (Ariadne Genomics, Rockville, MD, USA) and the GSEA tool [20] in Pathway Studio 11.0.5 [22] to identify over-represented biological processes within our differentially expressed dataset. As parameters for the GSEA method, we selected the Mann-Whitney U-test and a *p*-value threshold of 0.05.

### 2.6. High Resolution Fluorescent In Situ Hybridization

Mice were deeply anesthetized with ketamine-xylazine (140–148 mg/kg, i.p.) and sacrificed by intracardiac perfusion of PBS. Brains were dissected, frozen in isopentane at −25 °C and cut into 20 µm-thick frozen sections. FISH was performed following the RNAscope procedures (Advanced Cell Diagnostics (ACD) Newark, CA, USA) using a Cxcl10-specific probe (ACD reference 408921). Hybridization of a probe against the *Bacillus subtilis* dihydrodipicolinate reductase (dapB) (ACD reference 310043) gene was used as negative control. At least three independent experiments were performed and imaged.

### 2.7. Vessel Density Calculation

Experiments were performed on three months-old Cx43FL and Cx43KO brains (three animals per genotype). 20 µm-thick cryosections were post-fixed in PBS/PFA 4%, permeabilized in blocking solution (PBS/Triton X-100 0.25%/NGS 5%) followed by overnight incubation with Pecam-1/CD31 antibody (1/200 in blocking solution). Alexa594-coupled secondary antibodies were used to reveal Pecam-1/CD31 staining. Slices were imaged using a Leica DMRB wide-field microscope (10× objective, Leica Camera AG, Wetzlar, Germany) (three slices per animals). Pecam-1 staining from the parenchyma of cortex or hippocampus (meninges were excluded) was extracted from the images using ImageJ software (University of Madison, Madison, WI, USA).

Meninges were excluded. Number and size of vessels portions were obtained (Analyze particle, circularity = 0–0.8, size = 2 µm-Inf). Student T-test was used to determine differences between genotypes.

## 3. Results

### 3.1. Characterization of the Ribosome-Bound Transcriptome in Cx43-Deleted Hippocampal Astrocytes

We aimed to characterize how Cx43 deletion influences astrocyte functions. To do so, we focused on the transcriptional profile of Cx43-deleted astrocytes in astroglial Cx43 knock-out mice (Cx43KO) [14]. Astrocyte isolation and culture from neonates greatly modifies their molecular profiles even in absence of any pathogenic trigger [5]. To isolate the astrocyte “native” transcriptome, we developed an astrocyte-specific translating ribosome affinity purification (TRAP) approach [18], which allows focusing on mRNAs bound to ribosomes, thus potentially actively translated. To extract astrocyte polysomes in Cx43KO and floxed (FL) mice, we crossed them with Aldh1l1-eGFP/Rpl10a (Aldh1l1:L10a-eGFP) strain, in which the ribosomal protein L10a is fused with the eGFP and expressed under the control of the Aldh1l1 astrocyte-specific promoter [16]. Since astrocytes display regional molecular heterogeneity [21], we restrained our analysis to the hippocampus, where astrocyte Cx43 contribution to the brain physiology has been well studied [23], and performed extractions in 6 weeks-old tissues, when brain immune recruitment is at its maximum in Cx43KO [13]. GFP-tagged polysomes were immunoprecipitated in Aldh1l1:L10a-eGFP Cx43KO and in Aldh1l1:L10a-eGFP Cx43FL (control littermates) and mRNAs were extracted (Figure 1A). Three independent libraries for each genotype were generated and sequenced (Figure 1A). Importantly, known astrocyte-specific transcripts such as Gfap, Slc1a3 or Aqp4 were present in large quantity while most transcripts enriched or specific to myeloid cells (ex: Cx3Cr1), endothelial cells (ex: Cldn5), neurons (ex: Npas4), or oligodendrocytes (Cspg4) were absent, indicating the specificity of our approach [24]. In total, 118 genes were upregulated and 25 downregulated in Cx43KO compared to Cx43FL (Table 1 and Appendix A). We next determined if these molecular events fell in specific biological pathways. Strikingly, analysis of gene ontology (GO) profiles revealed that 1/4 of the significantly changed pathways in Cx43KO astrocytes belonged to immune responses (57 pathways) followed by cell signaling (33 pathways) and ionic homeostasis (25 pathways) (Figure 1B,C) (Appendix A). Accordingly, the most upregulated transcripts in Cx43KO astrocytes were all encoding inflammation-related proteins (Table 1) such as the chemokines Ccl5 and Cxcl10 and components of the complement activation such as C3ar1 or C1qb. To validate the expression of chemokines by Cx43KO astrocytes, we performed an in situ hybridization detection of Cxcl10 mRNA on hippocampus sections (Figure 2). Astrocytes were identified by the co-immunolabelling of GFAP. In Cx43FL control mice, no signal could be detected. In contrast, Cxcl10 FISH signals could be detected in some Cx43KO astrocytes of P25 and 6 weeks-old mice. While these results suggested that Cx43KO astrocytes adopted a pro-inflammatory profile, most canonical panreactive astrocyte markers such as Lcn2 (Lipocalin-2), Vim (Vimentin) or Gfap were not upregulated in Cx43KO astrocytes [5,6]. In addition, several anti-inflammatory markers were upregulated, among which the complement decay-accelerating factor CD55 and the membrane receptor CD274 which inhibits T-cell proliferation by blocking cell cycle progression and cytokine production [25] (Table 1). The Toll-like receptor Tlr4 was also downregulated in Cx43KO astrocytes (Appendix A), thus possibly reducing astrocyte capacity to activate an innate immune response [26]. Interestingly, these anti-inflammatory events seemed to be a specific signature of Cx43KO astrocytes as, except Serpina 3n, they were not observed in another recently described reactive astrocyte model of spinal cord injury (SCI), in contrast to most pro-inflammatory markers [7] (https://astrocyte.rnaseq.sofroniewlab.neurobio.ucla.edu) (Table 1). Altogether, these results strongly suggest that Cx43KO astrocytes display an atypical reactive profile with both pro- and anti-inflammatory properties.

### 3.2. Cx43KO Astrocyte Reactive Profile Is Not Related to Classical Signaling Inflammatory Pathways

Multiple intracellular signaling pathways have been associated with astrocyte reactivity such as the Jak/Stat (mainly via STAT1 and STAT3) or the NFκB pathways [27,28,29]. Interestingly none of the transcription factors related to these pathways were upregulated in Cx43KO (Table 1 and Appendix A), suggesting that if involved, their activation might be related to post-translational modification. We tested the phosphorylated status of STAT1 (Y701 phosphorylation) and NFκBp65 (Thr435 phosphorylation) on Western-blot of Cx43FL and Cx43KO hippocampus proteins extracted at 6 weeks (*n* = 3). No difference could be found in either case between Cx43FL and Cx43KO (Figure 3A,C). Next, we addressed the possible implication of the STAT3 pathway. To do so, we took advantage of the already described transcriptional profile of reactive astrocytes in mice submitted to spinal cord injury (SCI) either wild type mice or deleted for STAT3 (Table 1) [7,30] (https://astrocyte.rnaseq.sofroniewlab.neurobio.ucla.edu) and compared to Cx43KO astrocytes. This analysis showed that except for Serpina3n, most inflammatory markers upregulated in Cx43KO astrocytes were upregulated upon SCI either in WT and STAT3 deleted mice. Thus, activation of these transcriptional events would not be mediated by STAT3. Finally, since several genes upregulated in Cx43KO astrocytes such as Cxcl10, Ccl5, Igtp and Ilgp are expression targets of interferon gamma (IFNγ) [28,29], we compared the level of this cytokine on Western-blot of hippocampal proteins extracted in 6 weeks-old Cx43KO and Cx43FL mice (*n* = 3). This analysis showed no significant increase of IFNγ between Cx43KO and Cx43FL (Figure 3B,C).These results suggest that the reactive profile of Cx43-deleted astrocytes is not mediated by a classical inflammatory pathway.

### 3.3. Microvascular Density Is Increased in Cx43KO Brain

Several pro-angiogenic markers were upregulated by Cx43KO astrocytes, such as the Netrin receptor Unc5b [31] and the leucine-rich repeats and immunoglobulin-like domains 2 protein Lrig2 [32]. We thus hypothesized that an angiogenic activity might be increased in Cx43KO brains and could modify the brain vascular network. We evaluated the vascular density in 3 months-old Cx43KO brain compared to Cx43FL. Vessels were immunostained for the endothelial-specific membrane protein Pecam-1/CD31 on brain sections. Density of vessels was assessed calculating the number of vessel portions per µm^2^ of parenchyma in the hippocampus and the cortex (*n* = 3 animals per genotype, 3 slices per animal) (Figure 4). In both brain areas, this density was statistically increased in Cx43KO, indicating the presence of more vessels. In contrast, the mean size of vessels was significantly reduced in Cx43KO, suggesting that the additional blood vessels were smaller in diameter. Altogether, this advocates for the activation of angiogenic processes in Cx43KO mice, leading to the development of additional small vessels in the hipoccampus and cortex.

## 4. Discussion

How the cross talk between the brain and the immune system is regulated is a question of prime importance. Indeed, most neuropathologies from trauma to neurodegenerative diseases involve inflammatory responses in the brain. In these processes, astrocytes, which interface the brain vascular system and the CSF, have been shown to be crucially involved, modifying their properties and functions to adopt a diversity of reactive profiles to restrain or promote inflammation [5,6,7,33,34]. While many efforts have been put to characterize astrocyte reactivity in pathological conditions, how astrocytes influence the quiescent homeostasis between the immune system in the healthy brain is an open question. In a recent work, we demonstrated that Cx43, a gap junction protein enriched at the GL level, regulates this equilibrium [13]. In mice specifically deleted for astroglial Cx43, we observed a modest but constant immune recruitment of T- and B-lymphocytes, neutrophils and plasmocytes, and the development of a specific autoimmune response against Vwa5a, an extracellular matrix protein expressed by astrocytes. Surprisingly, while such immune reactions are normally toxic to the brain, they did not provoke any brain lesion in Cx43KO mice. To understand the molecular basis of such regulation, we analyzed here the polysomal transcriptome of Cx43KO astrocytes.

Most of the differential events in Cx43KO astrocytes belonged to inflammatory processes indicating that astrocytes were reactive in absence of Cx43. However, their profile did not fall into the two poles of reactivity defined by recent gene profiling studies, namely the A1 toxic and inflammatory phenotype mediated by NFκB, and A2 anti-inflammatory phenotype mediated by STAT3 [6,7]. Indeed, both inflammatory and anti-inflammatory molecules were upregulated in Cx43KO astrocytes indicated a mixed reactivity state. Cx43KO astrocytes lacked most pan-reactive astrocyte molecular events, such as the increase of GFAP, Vim (Vimentin) or Lcn2 (Lipocalin-2) [5,6,7,33,34,35]. Moreover, no activation of STAT1 and NFkB signaling pathways classically associated with astrocyte reactivity was detected, although their levels were measured here in whole tissues and not on purified astrocytes. Comparison of our data with the astroglial transcriptome of SCI WT or STAT3-deleted mice suggested that STAT3 activation may also not be involved in Cx43KO [7,30]. Interestingly, the unique astroglial transcriptomic profile nicely matched with the previously observed phenotype of Cx43KO mice [13]. First, the increase of Ccl5 and Cxcl10 had been detected in Cx43KO whole brains. Our present study indicated that Cx43KO astrocytes are the source of these chemokines. In contrast, the previously reported overexpression of Cxcl12 was not found here and might thus belong to another cell type. Second, while T-cells are constantly recruited in Cx43KO brains, no T-cell proliferation could be previously detected [13]. This observation is coherent with the overexpression of CD274 by Cx43KO astrocytes, a protein involved in the regulation of T-cell activation [36]. Third, the complement cascade is not activated in Cx43KO mice [13]. Consistently, the complement decay factor CD55 was upregulated in Cx43KO astrocytes [37]. Finally, we previously reported that shear stress applied to the vascular system was able to open the BBB in adult Cx43KO compared to Cx43FL, indicating a progressive BBB weakening in Cx43KO mice [13]. Here, we showed the possible activation of an angiogenic program in Cx43KO astrocytes with the upregulation of markers such as Lrig2. Consistently, an increased microvascular density was observed in Cx43KO brain. Interestingly, Angiogenetic processes are commonly described as detrimental to the BBB integrity [38]. Thus, the increased vascular density observed in Cx43KO might modify the brain vascular network and potentially affect BBB integrity.

Besides these inflammatory processes, we observed that Cx43KO astrocytes overexpressed other types of molecules that might participate to anti-inflammatory processes: (1) Adhesion molecules, and in particular several Claudins that form tight junctions (TJ). In normal healthy conditions, astrocyte endfeet forming the GL are devoid of TJs [39]. However, a recent report indicated that inflammation either mediated by IL1β or in an EAE model induces the expression of TJ proteins at the GL level, strengthening the GL barrier against immune cell penetration; (2) Proteins implicated in β-amyloïd clearance by astrocytes [40,41] suggesting a possible protective effect against Alzheimer disease as recently reported [42]; (3) Ion transporters which could compensate for ionic homeostasis modification induced by Cx43 gap junctions and hemichannels depletion.

How the modified brain immunity in absence of astroglial Cx43 influences physiopathological processes needs now to be further addressed. Interestingly, deletion of Cx43 in astrocytes was associated with a weaker inflammatory response upon sepsis [43] and an attenuated neuronal death in an Alzheimer disease model [42] and in a mouse model of middle cerebral artery occlusion [44]. The chronic immune recruitment observed in Cx43KO could possibly result in an enhanced innate immunity and allow the brain to resolve inflammation more efficiently.

## 5. Conclusions

We show here that Cx43, a gap junction protein enriched in astroglial perivascular endfeet, controls the immunoregulatory phenotype of astrocytes. When deleted for Cx43, astrocytes develop an atypical reactive profile, overexpressing inflammatory and anti-inflammatory molecules that may contribute to the constant but non-deleterious immune recruitment observed in the parenchyma, as well as to the activation of an angiogenic program. Cx43 is, therefore, a key regulator of the brain immune system homeostasis at the gliovascular interface.

## Figures and Tables

**Figure 1 brainsci-08-00050-f001:**
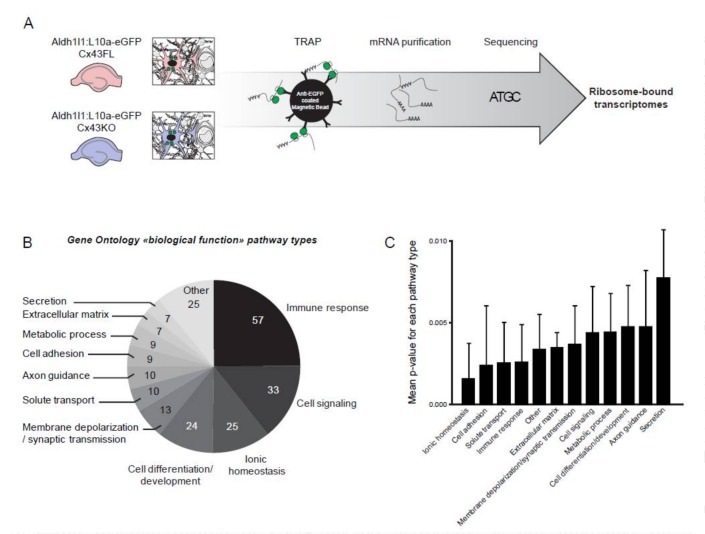
Comparative analysis of the polysomal astroglial transcriptome in Cx43KO and Cx43FL hippocampus; (**A**) Flowchart of the transcriptome procedure. Astrocyte polysomes were immunoprecipitated by TRAP from 6 weeks-old Aldh1l1:L10a-eGFP Cx43FL and Aldh1l1:L10a-eGFP Cx43KO hippocampi. Purified mRNAs were analyzed by RNASeq; (**B**,**C**) Gene Ontology “biological processes” analysis of differentially-expressed pathways between Cx43FL and Cx43KO astrocytes. (**B**) Circular diagram of pathway types, with the number of pathways for each type. (**C**) Detailed *p*-values of the Mann Whitney test for a change in expression between Cx43FL and Cx43KO astrocytes for each type of pathway.

**Figure 2 brainsci-08-00050-f002:**
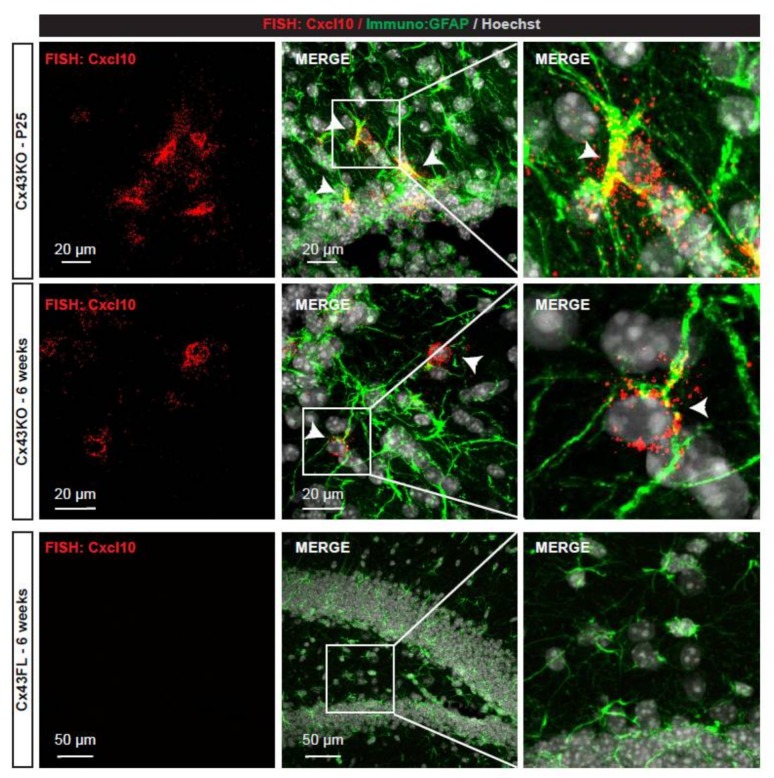
Detection of Cxcl10 mRNAs in Cx43KO hippocampal astrocytes; Confocal images of FISH detection of Cxcl10 mRNAs (red dots) on P25 and 6 weeks-old Cx43KO and 6 weeks-old Cx43FL hippocampal sections. Astrocyte processes were immunolabeled for the GFAP (green). Nuclei were stained with Hoechst (white). Right panels are larger magnifications of squared areas in the merge image (middle panel). Arrowheads indicate Cxcl10-expressing astrocytes.

**Figure 3 brainsci-08-00050-f003:**
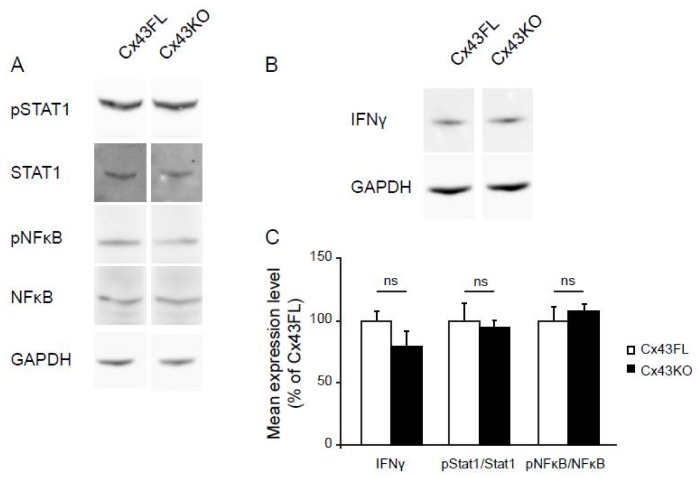
Comparative analysis of inflammatory-related signaling pathways in Cx43KO and Cx43FL hippocampus**.** Western Blot analysis of proteins extracted from 6 weeks-old Cx43FL (control) and Cx43KO hippocampi (*n* = 3). (**A**) Phosphorylation status of STAT1 and NFκB. Signals are normalized on the non-phosphorylated protein signal. GAPDH is used as a loading control; (**B**) IFNγ level. Signals are normalized on GAPDH; (**C**) Quantification of Cx43FL and Cx43KO signals in (**A**,**B**) ns stands for *p* > 0.05 (Mann-Whitney two-tailed test). Mean values are indicated ± the Standard Deviation (SD).

**Figure 4 brainsci-08-00050-f004:**
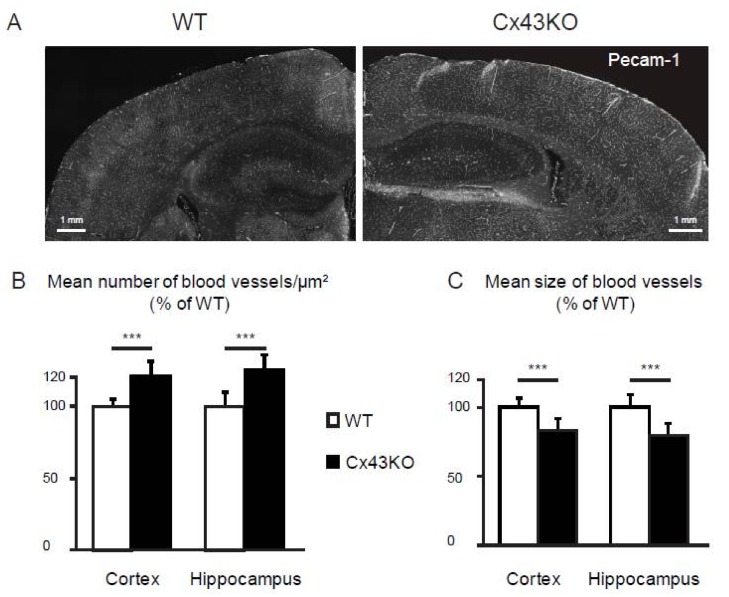
Comparative analysis of vascular density in the brain of Cx43FL and Cx43KO mice; (**A**) Representative images of Pecam-1 (CD31) immunostaining (white) on 3 months-old Cx43FL (control) and Cx43KO cortex and hippocampus sections; (**B**) Analysis of the number and size of blood vessels immnuolabeled for Pecam-1 in both genotypes. Student *T*-test, ***, *p* < 0.001.

**Table 1 brainsci-08-00050-t001:** List of the 50 most overexpressed ribosomal-bound transcripts in Cx43KO and Cx43FL 6-weeks old hippocampus. Transcripts in Cx43KO astrocytes are classified by function. The two columns on the right report the fold-change of these markers in astrocytes of a spinal cord injury (SCI) model compared to untreated adult astrocytes in WT and STAT3-deleted mice [7] *(https://astrocyte.rnaseq.sofroniewlab.neurobio.ucla.edu).* Transcriptional events common to Cx43KO astrocytes and SCI astrocytes are indicated in grey.

		Astrocyte Ribosome-Bound mRNAs (Hippocampus)				Astrocyte Total mRNAs (Spinal Cord)	
Gene Name	Molecule Name	Mean Reads Cx43FL	Mean Reads Cx43KO	Fold Change in Cx43KO Astrocytes	Padj	Fold Change in WT SCI Astrocytes	Fold Change in STAT3 KO SCI Astrocytes
	**Inflammation**						
Ccl5	chemokine (C-C motif) ligand 5	0.4	139.3	17.9	1.2 × 10^7^	9.6	10.34
Itgam	integrin alpha M	4.8	314.8	14.2	9.1 × 10^5^	7	8.02
C3ar1	complement component 3a receptor 1	0.8	63.2	13.5	6.7 × 10^4^	11.8	12.22
Cd34	CD34 antigen	2.8	140.4	13.1	4.3 × 10^4^	9	6.82
Cd53	chemokine (C-C motif) ligand 5	2.0	151.0	14.2	1.2 × 10^4^	7.4	7.5
Cspg4	chondroitin sulfate proteoglycan 4	4.1	127.9	11.4	7.2 × 10^3^	4.5	5.54
Lyz2	lysozyme 2	39.5	764.4	11.3	1.5 × 10^4^	17.8	16.52
Gbp4	guanylate binding protein 4	3.7	83.2	11.2	1.8 × 10^3^	10.34	19.5
Slfn5	schlafen 5	5.2	93.6	9.6	4.1 × 10^2^	14.24	12.1
Igtp	interferon gamma induced GTPase	66.6	706.6	9.7	1.6 × 10^7^	5.4	10.2
Iigp1	interferon inducible GTPase 1	54.8	803.7	9.5	2.1 × 10^2^	9.2	17.34
Psmb9	proteasome (prosome, macropain) subunit, beta type 9 (large multifunctional peptidase 2)	78.7	920.7	9.0	1.8 × 10^2^	5.98	9.12
C1qb	complement component 1, q subcomponent, beta polypeptide	35.2	328.6	8.3	2.9 × 10^2^	8.02	8.84
Scrt1	scratch homolog 1, zinc finger protein (Drosophila)	76.9	598.0	8.2	9.0 × 10^4^	ns	ns
Cxcl10	chemokine (C-X-C motif) ligand 10	80.8	663.5	8.1	1.4 × 10^2^	11.56	12.52
Fyb	FYN binding protein	27.0	185.8	7.6	9.8 × 10^3^	ns	10.08
Faah	fatty acid amide hydrolase	124.9	862.3	7.5	2.7 × 10^2^	6.4	ns
	**Anti-inflammation**						
Cd55	CD55 antigen	12.4	511.9	13.2	1.2 × 10^4^	ns	ns
Cd274	CD274 antigen	43.8	657.4	10.5	2.3 × 10^4^	ns	6.94
L1cam	L1 cell adhesion molecule	61.2	757.7	9.9	2.2 × 10^4^	ns	ns
Best3	bestrophin 3	7.4	110.9	9.6	1.7 × 10^2^	ns	ns
Serpina3n	serine (or cysteine) peptidase inhibitor, clade A, member 3N	202.7	1709.4	8.3	7.8 × 10^3^	8.14	ns
	Cellular junction						
Cldn3	claudin 3	0.0	60.2	15.2	2.8 × 10^4^	ns	ns
Cldn9	claudin 9	2.0	104.3	12.9	1.1 × 10^3^	ns	ns
Pcdhb18	protocadherin beta 18	27.4	249.1	8.7	2.7 × 10^3^	ns	ns
Cdh12	cadherin 12	75.2	579.6	7.8	2.1 × 10^2^	ns	ns
Unc5b	unc-5 homolog B (*C. elegans*)	17.3	264.5	9.8	1.1 × 10^2^	ns	ns
	**Angiogenesis**						
Lrig2	leucine-rich repeats and immunoglobulin-like domains 2	137.8	953.6	7.7	8.6 × 10^3^	ns	ns
Ccbe1	collagen and calcium binding EGF domains 1	49.8	394.3	8.1	6.3 × 10^3^	ns	ns
	**Glioma**						
Cmtm7	CKLF-like MARVEL transmembrane domain containing 7	3.6	66.2	9.9	2.4 × 10^2^	9.2	7.66
Rbl1	retinoblastoma-like 1 (p107)	21.3	266.9	9.7	1.5 × 10^3^	ns	ns
Cd109	CD109 antigen	4.0	58.1	9.2	3.4 × 10^2^	3.74	3.22
	**Ion Transport**						
Slc9a4	solute carrier family 9 (sodium/hydrogen exchanger), member 4	6.0	115.2	10.2	1.3 × 10^2^	ns	ns
Slc8a2	solute carrier family 8 (sodium/calcium exchanger), member 2	94.9	740.6	8.4	1.1 × 10^4^	ns	ns
Scnn1a	sodium channel, nonvoltage-gated 1 alpha	24.8	208.2	8.4	4.5 × 10^3^	ns	ns
Slc16a4	solute carrier family 16 (monocarboxylic acid transporters), member 4	42.7	397.4	8.3	2.6 × 10^2^	ns	ns
Cacna2d3	calcium channel, voltage-dependent, alpha2/delta subunit 3	56.8	407.6	7.8	1.0 × 10^2^	ns	ns
	**Aβ degradation**						
Mme	membrane metallo endopeptidase	6.0	141.3	11.8	1.2 × 10^4^	ns	ns
Pigz	phosphatidylinositol glycan anchor biosynthesis, class Z	6.4	122.5	10.3	1.2 × 10^2^	ns	ns

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
