# Peer review of "Connexin 43 Controls the Astrocyte Immunoregulatory Phenotype"

_brainsci, 2018, doi:10.3390/brainsci8040050_

Round 1

Reviewer 1 Report

This excellent work explores the function of Connexin 43 (Cx43) in the immunoregulatory phenotype of astrocytes. The transcriptome of hippocampal astrocytes deleted for Cx43 was analyzed. The astrocytes had an upregulation of molecules important for immune recruitment, complement activation, anti-inflammatory processes and angiogenesis. This work highlight the important role astrocytes play in many different processes in the central nervous system. The experiments were planned well and the results are sound. The conclusions drawn are comprehensible and explained well. The linguistic style is very good which makes the manuscript easy to read and understand. This manuscript will find the interest of many readers. Therefore I strongly advocate publication without further changes necessary.

Author Response

Comments and Suggestions for Authors

This excellent work explores the function of Connexin 43 (Cx43) in the immunoregulatory phenotype of astrocytes. The transcriptome of hippocampal astrocytes deleted for Cx43 was analyzed. The astrocytes had an upregulation of molecules important for immune recruitment, complement activation, anti-inflammatory processes and angiogenesis. This work highlight the important role astrocytes play in many different processes in the central nervous system. The experiments were planned well and the results are sound. The conclusions drawn are comprehensible and explained well. The linguistic style is very good which makes the manuscript easy to read and understand. This manuscript will find the interest of many readers. Therefore I strongly advocate publication without further changes necessary

We thank the reviewer for this very positive comments

Reviewer 2 Report

Boulay et al. investigated the immunoregulatory phenotype of astrocytes deleted for Connexin 43 (Cx43) with the analyses of the polysomal transcriptome. They found that in the absence of Cx43, astrocytes adopt an atypical reactive status with no change in astrogliosis markers, but with an upregulation of molecules promoting immune recruitment, associated with an increase in microvascular density in the brain. The study was well designed and the findings were clear, but some minor revisions are necessary to further clarify the importance of this study.

1. The present study should be a sequel to their previous studies. However, in Introduction, rather general and off-the-point descriptions were provided. The authors should accentuate much more the aim of this study and clarify the differences between the present study and their past studies.

2. Did they have any assumption that astroglial Cx43 is associated with any specific disease in the central nervous system? Further, is the downregulation of Cx43 in astrocytes related to any human brain disorders? They should specify these points in more detail.

3. In Figure 2, the images from the control mice (Cx43-wildtype) should be presented.

4. In Figure 3, why not present the blotting of pSTAT3 in Cx43KO and Cx43FL mice?. Also, the meaning of the data in Figure 3 is rather ambiguous since the analyzed protein derived from every constituent in the hippocampus. The authors should mention these points and the limit of the data.

5. In Figure 4, the size of blood vessels is hard to assess, and higher magnification images are necessary.

Author Response

Boulay et al. investigated the immunoregulatory phenotype of astrocytes deleted for Connexin 43 (Cx43) with the analyses of the polysomal transcriptome. They found that in the absence of Cx43, astrocytes adopt an atypical reactive status with no change in astrogliosis markers, but with an upregulation of molecules promoting immune recruitment, associated with an increase in microvascular density in the brain. The study was well designed and the findings were clear, but some minor revisions are necessary to further clarify the importance of this study.

We thank the reviewer for this very positive comments

1. The present study should be a sequel to their previous studies. However, in Introduction, rather general and off-the-point descriptions were provided. The authors should accentuate much more the aim of this study and clarify the differences between the present study and their past studies.

We have modified the introduction with an additional description of the previous study.

2. Did they have any assumption that astroglial Cx43 is associated with any specific disease in the central nervous system? Further, is the downregulation of Cx43 in astrocytes related to any human brain disorders? They should specify these points in more detail.

We are further discussing this point. Indeed, several studies have shown that astroglial deletion of Cx43 is beneficial to the brain and reduces inflammation. Are these effects related to the modified brain immunity described in our study has however not been addressed.

3. In Figure 2, the images from the control mice (Cx43-wildtype) should be presented.

We have added a picture of the control experiment.

4. In Figure 3, why not present the blotting of pSTAT3 in Cx43KO and Cx43FL mice?. Also, the meaning of the data in Figure 3 is rather ambiguous since the analyzed protein derived from every constituent in the hippocampus. The authors should mention these points and the limit of the data.

We agree with this comment. We have added a comment in the discussion about the fact that our Western blot analysis is  done on whole hippocampal tissues which might be a problem if signalization pathways are only activated in astrocytes. This is also precisely the reason why we took advantage of the STA 3 already published study which has been done specifically on astrocytes. The comparison shows that, apart from the inflammatory markers, all other transcriptional events found in Cx43KO astrocytes are probably not related to STAT3.

5. In Figure 4, the size of blood vessels is hard to assess, and higher magnification images are necessary.

A higher magnification of our images would not be more informative than the quantification already provided. Indeed, the differences between Cx43FL and FL are subtle. Provided images have a very good definition allowing the reader to zoom on the Pecam-1 labeling.